# Plasma Lipid Profiling Contributes to Untangle the Complexity of Moyamoya Arteriopathy

**DOI:** 10.3390/ijms222413410

**Published:** 2021-12-14

**Authors:** Michele Dei Cas, Tatiana Carrozzini, Giuliana Pollaci, Antonella Potenza, Sara Nava, Isabella Canavero, Francesca Tinelli, Gemma Gorla, Ignazio G. Vetrano, Francesco Acerbi, Paolo Ferroli, Elisa F. Ciceri, Silvia Esposito, Veronica Saletti, Emilio Ciusani, Aida Zulueta, Rita Paroni, Eugenio A. Parati, Riccardo Ghidoni, Anna Bersano, Laura Gatti

**Affiliations:** 1Clinical Biochemistry and Mass Spectrometry Lab, Department of Health Sciences, University of Milan, 20132 Milan, Italy; michele.deicas@unimi.it (M.D.C.); rita.paroni@unimi.it (R.P.); 2Laboratory of Neurobiology and UCV, Neurology IX Unit, Fondazione IRCCS Istituto Neurologico Carlo Besta, 20133 Milan, Italy; tatiana.carrozzini@istituto-besta.it (T.C.); giuliana.pollaci@istituto-besta.it (G.P.); antonella.potenza@istituto-besta.it (A.P.); sara.nava@istituto-besta.it (S.N.); isabella.canavero@istituto-besta.it (I.C.); francesca.tinelli@istituto-besta.it (F.T.); g.gorla10@campus.unimib.it (G.G.); eugenio.parati@istituto-besta.it (E.A.P.); anna.bersano@istituto-besta.it (A.B.); 3Neurosurgical Unit, Fondazione IRCCS Istituto Neurologico Carlo Besta, 20133 Milan, Italy; ignazio.vetrano@istituto-besta.it (I.G.V.); francesco.acerbi@istituto-besta.it (F.A.); paolo.ferroli@istituto-besta.it (P.F.); 4Diagnostic Imaging Department & Interventional Neuroradiology, Fondazione IRCCS Istituto Neurologico Carlo Besta, 20133 Milan, Italy; elisa.ciceri@istituto-besta.it; 5Developmental Neurology Unit, Fondazione IRCCS Istituto Neurologico Carlo Besta, 20133 Milan, Italy; silvia.esposito@istituto-besta.it (S.E.); veronica.saletti@istituto-besta.it (V.S.); 6Department of Diagnostic and Technology, Fondazione IRCCS Istituto Neurologico Carlo Besta, 20133 Milan, Italy; emilio.ciusani@istituto-besta.it; 7Neurorehabilitation Department, IRCCS Istituti Clinici Scientifici Maugeri, 20138 Milan, Italy; aida.zuluetamorales@icsmaugeri.it (A.Z.); riccardo.ghidoni@icsmaugeri.it (R.G.)

**Keywords:** moyamoya arteriopathy, lipidomics, angiogenesis, vasculogenesis, inflammation, RNF213, MMP-9, glycosphingolipids, sphingosine

## Abstract

Moyamoya arteriopathy (MA) is a rare cerebrovascular disorder characterized by ischemic/hemorrhagic strokes. The pathophysiology is unknown. A deregulation of vasculogenic/angiogenic/inflammatory pathways has been hypothesized as a possible pathophysiological mechanism. Since lipids are implicated in modulating neo-vascularization/angiogenesis and inflammation, their deregulation is potentially involved in MA. Our aim is to evaluate angiogenic/vasculogenic/inflammatory proteins and lipid profile in plasma of MA patients and control subjects (healthy donors HD or subjects with atherosclerotic cerebrovascular disease ACVD). Angiogenic and inflammatory protein levels were measured by ELISA and a complete lipidomic analysis was performed on plasma by mass spectrometry. ELISA showed a significant decrease for MMP-9 released in plasma of MA. The untargeted lipidomic analysis showed a cumulative depletion of lipid asset in plasma of MA as compared to HD. Specifically, a decrease in membrane complex glycosphingolipids peripherally circulating in MA plasma with respect to HD was observed, likely suggestive of cerebral cellular recruitment. The quantitative targeted approach demonstrated an increase in free sphingoid bases, likely associated with a deregulated angiogenesis. Our findings indicate that lipid signature could play a central role in MA and that a detailed biomarker profile may contribute to untangle the complex, and still obscure, pathogenesis of MA.

## 1. Introduction

Moyamoya arteriopathy (MA) is characterized by a progressive steno-occlusive lesion of the terminal part of the internal carotid arteries (ICAs), with a compensatory development of unstable collateral vessels at the base of the brain (*Moyamoya vessels*) [1,2]. These vascular hallmarks are responsible for recurrent ischemic and hemorrhagic strokes, leading patients affected by MA—often young adults and children—to severe neurological deficits, progressive physical disabilities, and even death [3,4,5,6,7]. MA is frequent in East Asian countries, while rarely reported in Caucasians. The association of MA with genetic disorders, the high familial rate, and the strong linkage with variants of *Ring Finger Protein 213 (RNF213*)/*Mysterin* coding gene in East Asian patients strengthen the role of genetic factors in MA pathogenesis [8,9,10,11,12,13,14,15]. Several reports implicated RNF213 as a sensor for mitochondrial dysfunction, hypoxia, and inflammation [14,16,17,18].

Nevertheless, the pathogenesis and etiology of MA remains unknown. It is believed that MA results from a complex mechanism in which acquired infectious, inflammatory, and flow dynamic conditions may trigger the disease in genetic susceptible individuals through angiogenic and vasculogenic pathway abnormalities [2]. The stenotic changes seen in MA are not characterized by lipid pools, inflammatory cells, or macrophage invasion to the sub-intimal layer, as typically seen in atherosclerosis [8]. Due to the detection of altered levels of endothelial progenitor cells (EPC), cytokines, chemokines, and growth factors in MA patient biological fluids, impaired angiogenesis and vasculogenesis have been invoked as potential disease mechanisms [2,9,19].

Lipidomics is the most powerful analytical tool to study lipids in biological specimens and their biochemical involvement in human diseases. The structure and the composition of lipids in cells and tissues can change in response to pathophysiological modifications. Endogenous lipids are not just the major constituents of cell membranes, involved in structural compartmentalization, energy storage, and signal transduction, but they play as key pathophysiological mediators regulating and fine-tuning several intracellular functions including apoptosis, proliferation, response to stress and inflammation [20,21]. Thus, due to these newly discovered functions and their pivotal role, they have gained the title of “bioactive lipids” [22]. Many lipids have been proposed to function as extracellular mediator molecules after being released via exocytosis or transporter-mediated pathways to the extracellular milieu [23]. Subsequently, the interaction with specific receptors launches downstream signaling cascades in target cells activating a plethora of different cell responses including migration, differentiation, survival, proliferation, and apoptosis. Lipid mediators hence can regulate a broad spectrum of events such as inflammation, immunity, and angiogenesis. Bioactive lipids are recognized as novel biomarkers that would contribute to innovative treatment methods, early intervention, and better patients’ prognosis. In particular, a significant effort was made by decoding the role of lipid in mediating and resolving inflammation [24,25,26]. Endogenous bioactive lipids are well acknowledged for their function in inflammatory processes: they regulate hypervascular reactivity, pain, leukocyte trafficking, and clearance, and also can contribute to originating, coordinating, resolving, and restricting inflammation itself [22,27,28,29,30]. Among all lipid classes, sphingolipids participate in numerous inflammatory processes and are responsible for controlling intracellular trafficking and signaling, cell growth, adhesion, vascularization, survival, and apoptosis. Ceramide and its metabolites seem to mediate the pro-inflammatory activities and cytokine production. However, ceramides have also been shown to negatively regulate some pro-inflammatory cytokines, suggesting a more complex role for them in inflammation. Ceramide and its metabolites are also important for preservation of the endothelial/vascular integrity and function, whereby alterations of these sphingolipids are associated with vascular dysfunctions [31,32].

Thus, identification of a lipidomic signature may be relevant to detect potential diagnostic, prognostic, and predictive biomarkers in various diseases [33]. Moreover, the study of the concentrations of a subset of lipids in human disease can shed light into cellular processes and unravel alterations in one or more pathways [34]. Lipids can be studied by targeted or untargeted lipidomics, both using high resolution Mass Spectrometry (MS). The untargeted approach aims to objectively identify and semi-quantify a very large number of all potential lipid species, thus giving an overall lipid fingerprint of the sample. A high-sensitive analysis dedicated to identifying and exactly quantifying only few specific classes of lipids is a targeted approach [35,36].

It is increasingly evident that the pathophysiology of MA encompasses many different mechanisms and that the finding of disease biomarkers will refine and improve our understanding of this arteriopathy. In this respect, protein and lipid plasma profiles may provide a useful combination of molecular and functional markers of the disease.

The aim of the present study is to address the question whether a detailed biomarker profile may contribute to untangle the complex, and still obscure, pathogenesis of MA. Thereby, we evaluated lipid, angiogenic and inflammatory factors released in plasma of a cohort of Italian MA patients.

## 2. Results

### 2.1. MA Patients, Healthy Donors, and Unrelated Subjects Recruitment

Among the original cohort of more than 150 patients of the GEN-O-MA study [37], 40 MA adult Caucasian patients in whom it was possible to collect whole blood samples were included in the present study (Appendix A). The full study methodology has been already reported elsewhere [19,37]. The selected patients displayed a mean age of 45.4 ± 12.4 years, with a prevalence of female subjects (82.5 %).

The disease presented with an ischemic event in 32.5% of patients, a hemorrhagic stroke in 20% of them, and a transient ischemic attack (TIA) in 27.5% of cases. In the remaining cases the disease was diagnosed for headache, trauma, or during investigations.

Thirty-five healthy donors (HD) were recruited as controls. They displayed a mean age of 41.6 ± 11.9 years, with a percentage of females = 91.4%. Additionally, 16 subjects (43.7% females) with a mean age of 53.4 ± 13.4 years and presenting unrelated atherosclerotic cerebrovascular diseases (ACVD) were selected as a further control group.

### 2.2. Reduced MMP-9 Level in Plasma of MA Patients

In order to find a possible pathophysiological mechanism, vasculogenic/angiogenic and inflammatory pathways were investigated through ELISA analysis, conducted on plasma collected from MA patients. A subgroup of MA patients was included, comparing them to HD and ACVD subjects. 

The level of selected proteins potentially released in plasma, respectively, angiogenic/vasculogenic (Angiopoietin-2, Ang-2; Metalloproteinase-9, MMP-9; Vascular Endothelial Growth Factor-A, VEGF-A) and inflammatory factors (chemokine (C-C motif) ligand 5, CCL5/RANTES; interleukin 8, IL-8; interleukin 6, IL-6), was measured. The results of the analysis are shown in Figure 1.

A lower level of MMP-9 was detected in plasma of MA patients in comparison with ACVD and HD subjects, being the difference between MA and ACVD statistically significant (Figure 1B). A slight, but not statistically significant, increase in inflammatory cytokines (IL-6 and IL-8) released in plasma of MA with respect to HD and ACVD was found.

### 2.3. Overall Lipid Content in Plasma of MA Patients: Untargeted Lipidomic Approach

The plasma lipid profile was evaluated in MA patients (n = 15, 100% females, mean age 45.7 ± 9.5) in comparison to HD (n = 15, 100% females, mean age 46.5 ± 9.8). Using an untargeted lipidomic approach, we identified n = 1959 different lipids in plasma of both MA and HD. The nomenclature of some of the modulated lipid species are included in Table 1. Performing a univariate statistical analysis to the entire dataset, MA patients showed a surprisingly lower content of lipids in plasma as compared to HD (*p* value < 0.05) (Figure 2A). MA patients were significantly depleted in plasma lipids belonging to the glycosphingolipid class, namely Hexosylceramide (HexCer) that is the sum of GlcCer and GalCer, undistinguishable in MS analysis, Gb3, Lactosylceramide (LacCer), as well as in phospholipid class such as Phosphatidylcholine (PC), Phosphatidylinositol (PI), and EtherPhosphatidylcholine (EtherPC) (Figure 2B).

The discriminant analysis (PLS-DA) even better elucidated the lipidomic profiles differences among the two groups and showed a separation of 23.1% on principal component (PC1). The PC1 represents the new dimension in which the initial variables are compressed and represents the maximum of the separation that can be reached within these clusters and variables (Figure 3A). The Variance Importance in Projection scores (VIP) derived from PLS-DA were used for ranking the discriminating features, taking a cut-off value >1.5. In Figure 3B, only lipids with a distinctive discriminant power (VIP > 1.5) are visualized by a heatmap and ordered according to their subclasses. Univariate analysis was performed to further validate this dataset and to test whether the trend in the plasma levels of the single discriminant lipids was consistent with this clusterization (Figure 3C). This last analysis led us to unravel a particular depletion in MA of some lipid species such as Cer 24:1, GM3 18:0, SULF 16:0 and many lipid classes, namely, HexCer, LacCer, Gb3, LPC, LPE, PC, PE, PI, EtherPC, EtherPE, and DAG. Most of these lipid variations overlap with those observed in the comparison of the overall lipid content (Figure 2).

### 2.4. Targeted Lipidomics Approach

A quantitative targeted LC-MS/MS analysis showed that the free sphingoid base profile was altered in plasma of MA patients in comparison to HD (Figure 4).

Specifically, Sph, DHSph, S1P, and DHS1P concentrations were augmented in MA patient plasma with respect to HD plasma values, being all differences statistically significant between the two patient sets.

## 3. Discussion

Although the pathologic role of RNF213/Mysterin in MA is little understood, recent studies highlighted its function as a metabolic gatekeeper with important roles in cellular response to hypoxia, vascular stability, and inflammation [17,38]. Different rare RNF213 variants have been reported in Caucasian cases, preferentially clustering in the E3 domain [11,39]. Examples of protein targets of conventional ubiquitination by RNF213 are master regulators of angiogenesis, like HIF-1α, and inflammation, as NFAT-1 [18,40]. Noticeably, RNF213 is also capable of influencing angiogenesis by inducing the NF-kB-mediated expression of inflammatory cytokines [14,17,38]. The chronic inflammation induces fibrosis or angiogenesis due to the disruption of the adaptive responses: hyperplasia of intimal VSMCs and neovascularization by proliferation of ECs cause lumen stenosis and collateral stenosis [41]. A scheme summarizing the implications of RNF213 in the vasculogenesis/angiogenesis/inflammatory pathways is shown in Appendix A.

Obesity and dyslipidemia have not been yet considered to be MA metabolic anomalies [42]. However, very recently it has been suggested a potential link between MA pathogenesis and lipid metabolism [43], although neither study showed direct association between *RNF213* mutations and dyslipidemia. An outline of the putative role of RNF213 in lipid metabolism in MA is presented in Appendix A.

Based on the controversial literature results and the lack of a consensus on reliable biomarkers for MA, we aimed here to uncover potential plasma biomarkers belonging to lipid metabolism, also for its potential connection with the angiogenic/vasculogenic and pro-inflammatory pathways. Specifically, we searched for putative MA plasma lipid biomarkers, because an alteration of selected lipid species could contribute to substantiate the typical disease hallmarks.

Among proteins released in plasma, we found a lower level of MMP-9 in MA patients in comparison to HD and ACVD subjects, as well as a slight—not statistically relevant—increase in pro-inflammatory cytokines (i.e., IL-8 and IL-6). MMP-9 has an important role in extracellular matrix (ECM) remodeling and regulation of angiogenesis. Indeed, MMP-9 is the main enzyme in the degradation of ECM proteins. The production and maintenance of the ECM is an important aspect of endothelial cell (EC) function. The ECM provides mechanical support to ECs and mediates signaling via secreted molecules. In the absence of angiogenesis stimulation, ECM helps to maintain ECs in a quiescent state. MMP-9 plays a major role when the ECM is degraded and the basement membrane is destroyed, thus causing ECs to migrate from existing blood vessels to newly formed blood vessels. Although it is not completely known which types of cells (ECs, VSMCs, EPC, or immune cells) form the primary lesion in MA, the histopathological features of this rare condition are intimal fibrous thickening and VSMC proliferation in the ICAs [42]. An irregular atrophy of VSMCs and a pathological increase in ECM of the tunica media have been described in large vessel damage [44]. Thus, the decreased MMP-9 expression found in plasma of our MA patients could explain the ECM increase observed in MA cerebral lesions, indicating a deregulated angiogenesis mechanism. As previously reported, MMP-9 exerts an important role also in the migration of VSMCs, necessary for the vessel formation. Thus, MMP-9 decrease in plasma could suggest a defective migration of VSMCs, leading to a dysfunction in the formation of new collateral vessels [45]. Another intriguing hypothesis is that MMP-9 decrease in MA patients represents the effect of a defective production of anti-inflammatory cytokines [41]. Interestingly, a recent study has shown that serum levels of MMP-9 could discriminate the phenotype of MA patients, since hemorrhagic MA patients had a higher level of MMP-9 than ischemic MA patients. Therefore, it has been hypothesized that the level of MMP-9 might serve as a biomarker for predicting hemorrhage in MA patients [46]. Previous studies have reported contrasting or uncertain results regarding the role of MMP-9 in MA [47,48,49]. Recently, Blecharz-Lang and co-authors did not find any modulation in serum MMP-9 of MA patients, although they observed that MMP-9 was overexpressed and secreted by EC cultures maintained with MA serum [47]. Evidence reported in the oldest studies were influenced by age-related heterogeneity both for MA patients and HD. Indeed, a group of MA pediatric patients exhibited significantly higher plasma concentrations of MMP-9 but only if compared to adult HD [48]. Fujimura and colleagues showed an increased expression of serum MMP-9 but in MA patients aged from 8 to 62 years [49].

By examining the plasma lipid profile, we found a surprisingly lower content of lipids in MA patients with respect to HD. The decreased levels of lipids that we observed in MA plasma when compared to HD, should likely be due to specific lipid classes, different from total cholesterol, LDL-cholesterol, and triglycerides which previously appeared within the normal limits in MA patients when compared to HD [50]. Since glycosphingolipids and phospholipids are plasma-membrane components, their reduced level in plasma could correlate with a decrease of cellular debris, due to a reduction of peripherally circulating progenitor cells [19], as an effect of the major cerebral recruitment of various cells (e.g., cEPCs, cVSMCs, immune cells) [51].

Overall, untargeted lipidomics approach suggested that the lipid profile of MA patients is dramatically homogeneous but deeply separated from that of HD subjects, as displayed by both discriminant and univariate analyses (see Figure 2 and Figure 3). Interestingly, we reported the results of a homogeneous group of patients that is highly representative of the whole cohort of our patients. All of them were women with mean age around 45y, bilateral involvement, and high disease severity, as assessed by Suzuki grading ≥ III (Appendix A). Given the tight homogeneity of our sample, we believe that our plasmatic lipid profile findings are likely associated with typical MA pathological features.

The lipids that were found altered using the discriminant analysis (score plot) of the lipidome, included ganglioside GM3 18:0, that is the predominant long-chain GM3 molecular species in human plasma. The decrease of this lipid species in MA patients is aligned with what observed by untargeted lipidomics for other glycosylated sphingolipids, as discussed before. Interestingly, serum GM3 was reported to play a role in innate immune function of macrophages, thus impacting angiogenesis. Specifically, long-chain GM3 (and not very-long-chain GM3) functions as an anti-inflammatory Toll-like receptor 4 (TLR4) modulator [52]. The same is true for sulfatide that was reported to exert an anti-inflammatory role by hindering the co-localization of TLR4 and lipid rafts [53]. Thus, the diminution of both GM3 18:0 and SULF 16:0 in MA patients appears to reflect the inflammatory feature of MA. A further intriguing result of the untargeted approach is the increase of cardiolipin (and specifically its 64:8 molecular species), a typical mitochondrial lipid, in MA plasma. This agrees with the mitochondrial abnormalities observed in MA circulating endothelial cells [54,55]. In MA patients, we also found a plasmatic depletion of long-chain Cer 24:1, which is synthetized by a specific Cer2 Synthase and has been previously reported to represent the major Cer species involved in defective angiogenesis [56]. Moreover, a low concentration of very-long-chain Cers has been reported to be related to increased cardiovascular risk [57]. The evidence that a depletion of very-long chain Cer 24:1 in MA associates with the increase of S1P and DHS1P, the most abundant sphingoid bases in plasma (see targeted analysis reported in Figure 4), may reflect a release of pro-angiogenic and cell growth-inducing lipid biomarkers from the newly produced MA vessels. Indeed, phosphorylated sphingoid bases and ceramides are well known to display opposite functions. Specifically, S1P and DHS1P are pro-angiogenic/proliferative factors, whereas ceramide has a pro-apoptotic role [58].

In order to help the interpretation of the potential origin of the reported plasma lipid and protein changes in a pathologically meaningful context, we mapped the deregulated species as well as the predicted associated effects (Figure 5).

Many recent studies have re-evaluated the importance of lipid metabolism for MA and new intriguing features of such poorly understood RNF213 protein have been suggested [17]. In addition to a clear direct role of RNF213 in triglyceride accumulation in lipid droplets [43], unbiased analyses revealed RNF213 as an important modulator of lipotoxicity caused by saturated fatty acids through inhibition of stearoyl-CoA desaturase (SCD-1) [16]. Ge et al. reported that the HDL-cholesterol level was inversely associated with the risk of MA [59]. Moreover, dyslipidemia was shown as a risk factor for contralateral progression in patients with unilateral MA [60]. Hirano et al. also reported that dyslipidemia was associated with symptomization of asymptomatic MA patients [61]. In spite of such evidence, the baseline plasmatic levels of total cholesterol, LDL-cholesterol, and triglycerides seem to be all within the normal limits in MA patients when compared to HD [50].

In conclusion, targeting lipid metabolism may be a potential therapeutic strategy against some pathological hallmarks of MA. However, the feasibility of modulating lipid metabolic processes as a multi-target treatment of MA requires precise mechanistic control, including disease timing, cell specificity and host lipidome profiles to achieve a holistic precision medicine approach.

There are several limitations to our study: firstly, the sample size of MA patients and controls is relatively small, after patient selection based on ethnicity, age, and clinical data. Thus, our findings should be verified in a larger cohort and in other biological samples (e.g., cerebrospinal fluid). Moreover, the study could suffer from selection bias since a strict sex- and age-matched control matching was not fully respected. The validation of our results on a larger population and the correlation with clinical data could help our understanding of their role in MA, leading to the discovery of reliable biomarkers and the identification of therapeutic targets.

## 4. Materials and Methods

### 4.1. Moyamoya Patients and Healthy/ACVD Controls: Inclusion Criteria

This was an observational study conducted on MA patients, diagnosed following the literature criteria [62], belonging to the GEN-O-MA study. The full methodology of the study is reported elsewhere [37]. From the original population of 150 patients consecutively enrolled at the Neurology IX Unit of the Fondazione IRCCS Istituto Neurologico “C. Besta” (Milan), between November 2014 and July 2021, a patient subgroup was selected for the present study (see Appendix A for clinical–demographical characteristics). Individuals fasted within 12 h, and subjects with endometriosis and/or positive for HIV, HBV, or HCV were excluded from this study.

A population of age- and sex-matched Caucasian healthy donors (HD) recruited from the general population was collected as control. Since vascular risk factors could influence cEPC populations, HD were interviewed for their medical history, and subjects presenting at least one of the following parameters out of normality ranges were excluded: blood pressure, glycemia, and cholesterol level [63,64,65,66]. Smokers, subjects with active duodenal or gastric ulcer, subjects that had undergone drug treatments in the preceding 48 h, or subjects with present or previous neoplastic, infectious, inflammatory, or cardiovascular diseases were also excluded.

Additionally, a group of age- and sex-matched Caucasian atherosclerotic cerebrovascular disease (ACVD) patients were recruited as further controls. ACVD subjects were diagnosed when patients had either an internal carotid, middle cerebral artery, cerebral anterior artery occlusion, or stenosis from atherosclerotic origin. They underwent conventional catheter digital subtraction angiography and morphological imaging by MRI as well.

### 4.2. Ethical Issues

The study design was approved by the Ethics Committee of the Fondazione IRCCS Istituto Neurologico “C. Besta” of Milan (report no. 12, 10/01/2014) and was performed in accordance with the 2013 WMA Declaration of Helsinki. Since it was designed as a pure observational study, patients underwent diagnostic procedures and received therapy according to local practice. Informed written consent for study participation and sample collection from all patients and controls were mandatory for study inclusion. Privacy procedures were applied to protect patients’ and healthy controls’ personal identities.

### 4.3. Blood and Plasma Samples Collection

Twenty-four milliliters of peripheral blood were withdrawn by venipuncture from MA, HD, and ACVD subjects and collected in tubes containing ethylenediaminetetraacetic acid (EDTA) as anticoagulant (Vacuette^®^, Preanalitica s.r.l., Caravaggio, Italy).

One vacutainer was stored at −20 °C for future molecular analysis. For plasma collection, two vacutainers were centrifuged for 10 min at 300× *g*; plasma was transferred into a new tube (SARSTEDT AG and Co, Nümbrecht, Germany) and stored in aliquots at −80 °C until use.

### 4.4. Clinical–Radiological Factors

For all patients, demographic and clinical features were collected applying a standardized form [37].

MA was classified into the bilateral or unilateral types depending on the number of distal ICAs involved, as observed on conventional angiography [62]. Diagnosis of ischemic or hemorrhagic stroke was confirmed by conventional neuroimaging (computerized tomography scan and magnetic resonance imaging). The MA severity was assessed by Suzuki scale [67].

### 4.5. ELISA

ANG-2 (Boster Biological Tecnhology Co., LTD, Pleasanton, CA, USA), MMP-9 (ThermoFisher, Monza, Italy), CCL5/RANTES (ThermoFisher, Monza, Italy), VEGF-A (Boster Biological Tecnhology Co., LTD, Pleasanton, CA, USA), IL-8/CXCL8 (ThermoFisher, Monza, Italy), and IL-6 (ThermoFisher, Monza, Italy) concentrations were assessed using highly sensitive enzyme-linked immunosorbent assay kit in triplicate on plasma samples. Enzyme-linked immunosorbent assay was performed according to the manufacturer’s instructions. 

### 4.6. Chemicals and Reagents for Lipidomics

The chemicals acetonitrile, 2-propanol, methanol, chloroform, formic acid, ammonium acetate, ammonium formate, and dibutylhydroxytoluene (BHT) were purchased from Sigma-Aldrich (St. Louis, MO, USA). All aqueous solutions were prepared using purified water at a Milli-Q grade (Burlington, MA, USA). 

### 4.7. Untargeted Lipidomics

Lipids from plasma (25 µL) were diluted with water (1:4 serum, 1:2 CSF, *v*/*v*), extracted by a mixture of methanol/chloroform (850 µL, 2:1, *v*/*v*) and analyzed by LC-MS/MS consisting of a Shimadzu UPLC coupled with a Triple TOF 6600 Sciex (Concord, ON, CA) [68]. All samples were analyzed in duplicate in positive electrospray ionization. Spectra were contemporarily acquired by full-mass scan from *m/z* 200–1500 and top-20 data-dependent acquisition from *m/z* 50–1500. Declustering potential was fixed to 50 eV, and the collision energy was 35 ± 15 eV. The chromatographic separation was reached on a reverse-phase Acquity CSH C18 column 1.7 μm, 2.1 × 100 mm (Waters, Franklin, MA, USA) by a gradient between (A) water/acetonitrile (60:40) and (B) 2-propanol/acetonitrile (90:10), both containing 10 mM ammonium acetate and 0.1% of formic acid. 

### 4.8. LC-HR-MS Data Processing 

The spectra deconvolution, peak alignment, and sample normalization were attained using MS-DIAL (ver. 4.0, RIKEN Center for Sustainable Resource Science, Yokohama, Japan, http://prime.psc.riken.jp/compms/msdial/main.html, accessed on 29 December 2020). MS and MS/MS tolerance for peak profile was set to 0.01 and 0.05 Da, respectively. Identification was achieved matching spectra with LipidBlast database or in-house built mass spectral library. Intensities of analytes were normalized by Lowless algorithm and those with a CV% superior to 30% in the QC pool sample were excluded. The lipid classes considered were acylcarnitines (ACar), cholesterol (chol), cholesterol esters (CE), dihydroceramides (DHCer), ceramides (Cer), hexosylceramides (HexCer), lactosylceramides (LacCer), globotriaosylceramide (Gb3), sulfatides (SULF), gangliosides (GM3), sphingomyelins (SM), lysophosphatidylcholines (LPC), lysophosphatidylethanolamines (LPE), phosphatidylcholines (PC), phosphatidylethanolamine (PE), phosphatidylinositoles (PI), and plasmalogens—that are ether linked phosphatidylcholines (EtherPC) and vinyl linked phosphatidylethanolamines (EtherPE). The lipids are indicated throughout the paper by their total number of carbon atoms and degree of unsaturation (i.e., PC 40:2) or eventually specifying the acyl chains detected (i.e., PC 18:0/18:1).

### 4.9. Sphingoid Long-Chain Bases Sphingoid Long-Chain Bases Determination 

Sphingolipid extraction was performed as already described in the previous section (par 4.7) except for the addition of alkaline methanolysis step to enhance the recovery of low abundant sphingolipid species. The clear supernatant was injected on a Shimadzu UPLC coupled with a Triple TOF 6600 Sciex (Concord, ON, CA) equipped with Turbo Spray IonDrive. Sphingosines analysis was completed on Acquity BEH C18 column 1.7 μm, 2.1 × 100 mm (Waters, Franklin, MA, USA) by using, as mobile phase (A) 0.2% formic acid 2 mM ammonium formate in water and as mobile phase (B) methanol 0.2% formic acid 1 mM ammonium formate. The flow rate was 0.3 mL/min and the column temperature was 30°C. The elution gradient (%B) was set as follows: 0–10 min (80–99%), 10–15 min (99%), 15–15.2 min (99–80%), 15.2–20 min (80%). Sphingoid long-chain bases were determined by monitoring the high-resolution transitions *m*/*z* 380.25 > 264.26 (S1P), 300.28 > 282.27 (Sph), 302.30 > 284.29 (DhSph), and 382.27 > 284.29 (DhS1P), by applying a DP of 50 eV and CE 30 ± 15 eV. Quantitative analysis was corrected for internal standard responses (sphinganine d17:0; *m*/*z* 288.28 > 270.27), interpolation with calibration curves and the results were expressed as µM.

### 4.10. Statistics and Data Visualization 

#### 4.10.1. ELISA Statistical Analyses

Data were expressed as mean ± standard deviation, and statistical significance (* *p* < 0.05) was calculated through Student’s *t*-test by using GraphPad Prism 8 software (GraphPad Software; San Diego, CA, USA). 

#### 4.10.2. Lipidomics Statistical Analyses

Graphs and statistical analyses were prepared with GraphPad Prism 7.0 (GraphPad Software, Inc., La Jolla, CA, USA), and with MetaboAnalyst 4.0 (ver. 4.0, McGill Data Center and Compute Canada, Montréal, Canada, https://www.metaboanalyst.ca, accessed on 17 February 2021). Data were checked for integrity, filtered by interquartile range, log-transformed, and auto-scaled. Univariate statistical analysis was performed using unpaired t-test for two-group comparison. For multivariate analysis partial least squares discriminant analysis (PLS-DA) was performed in order to increase the group separation and investigate the variables with a high Variance Importance in Projection score (VIP > 1.5). *p* < 0.05 was considered statistically significant. Data are shown as mean ± SD.

## 5. Conclusions

The specific histopathological features of MA are well known and are represented by an excessive but defective neovascularization which may cause intimal thickening and occlusion and further compensatory, atypical collateral formation. It is also increasingly evident that angiogenic/vasculogenic factors and pro/anti-inflammatory cytokines appear to be strongly correlated with MA features.

In this study, we contributed to untangle the complexity of MA pathogenesis thanks to a novel approach that analyzes the composition of plasma samples from MA patients in terms of both peptides and lipids. In particular, our findings indicate that the plasma lipid profile of MA patients is definitely peculiar, thus highlighting a novel source of reliable clinically useful biomarkers for the disease.

## Figures and Tables

**Figure 1 ijms-22-13410-f001:**
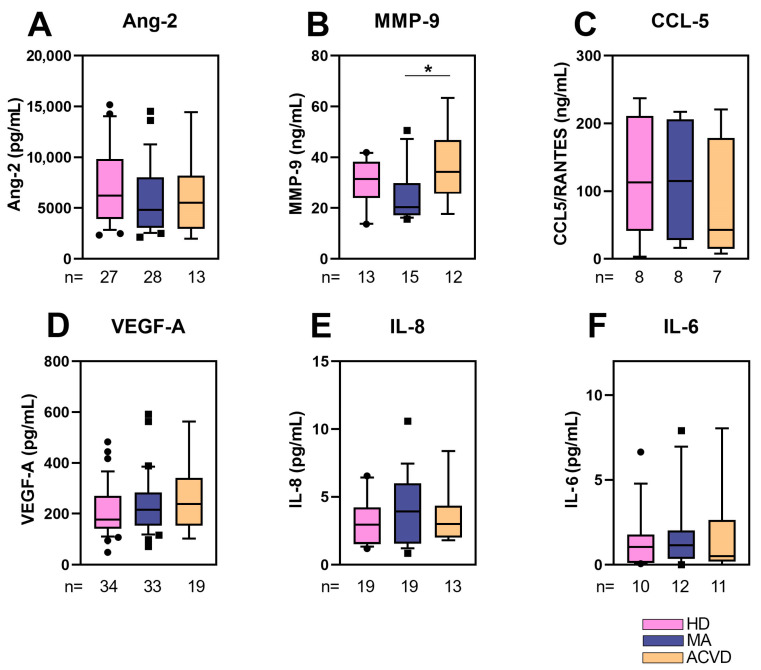
(**A**) Angiopoietin-2 (Ang-2), (**B**) matrix metalloprotease-9 (MMP-9), (**C**) chemokine (C-C motif) ligand 5 (CCL5/RANTES), (**D**) Vascular Endothelial Growth Factor-A (VEGF-A), (**E**) interleukin 8 (IL-8/CXCL8), and (**F**) interleukin 6 (IL-6) concentration (pg/mL for **A**,**D**,**E**,**F** or ng/mL for **B**,**C**) in plasma collected from MA patients, HD and ACVD subjects. The boxes represent data obtained in the range 25th–75th percentile; the line across the boxes indicates the median value; the lines above and below the boxes indicate extreme values (10th or 90th percentile). The statistical significance (* *p* < 0.05) was calculated through Student’s *t*-test. Values of at least three independent experiments are shown.

**Figure 2 ijms-22-13410-f002:**
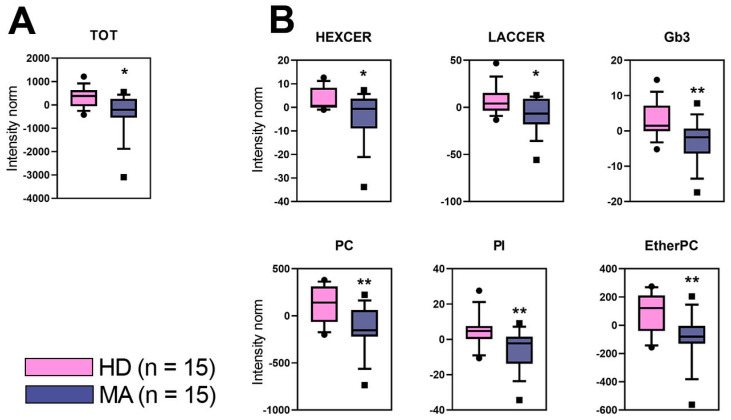
(**A**) Total lipid content in plasma of MA patients (MA, n = 15) and of healthy donors (HD, n = 15). (**B**) Lipid classes significantly decreased in plasma of MA patients. Other lipid classes were not visualized since their alteration does not reach any statistical significance. In this analysis, all lipids identified in the same class were summed and included in the boxplots. Data were reported as log-transformed and auto-scaled mass intensities (Intensity norm). Statistical significance was evaluated by unpaired *t*-test. *p* values are schematized as follows: * < 0.05; ** < 0.01.

**Figure 3 ijms-22-13410-f003:**
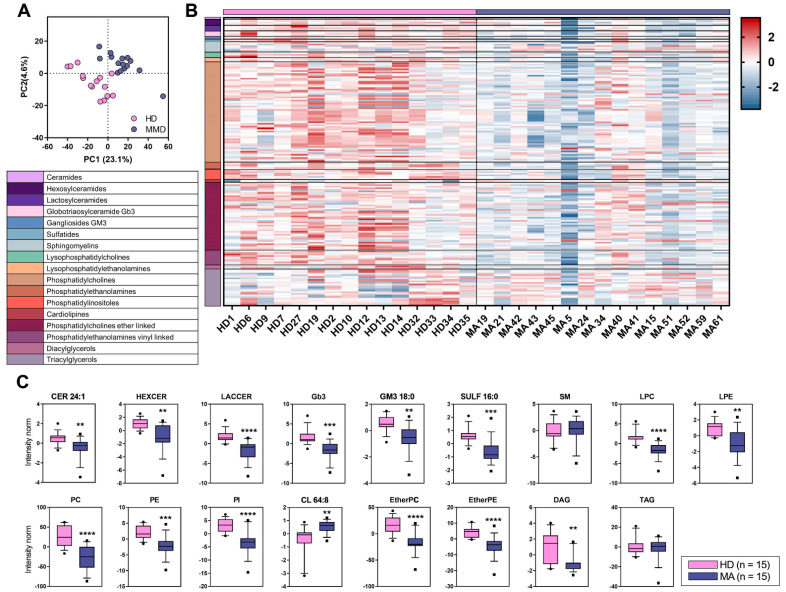
Plasma lipid alteration in MA patients (n = 15) in comparison to age and sex matched healthy donors HD (n = 15). (**A**) Discriminant analysis (score plot) of the lipidome in HD and MA patients. The axes are ranked according to their importance in the group discrimination. In the x-axis, component 1 (PC1, 23.1%) represents the maximum of the separation that can be reached within these clusters and variables, whereas, in the y-axis, component 2 (PC2, 4.6%) represents the direction that contains the most remaining variance. (**B**) Heatmap of the lipids highly correlated with the disease (n = 175), chosen within those with a Variance Importance in Projection (VIP) score >1.5, ordered by lipid classes, coded by different colors. The concentrations were autoscaled and log-transformed for visualization. The color-scale differentiates values as high (red), average (white), and low (blue). (**C**) Boxplots represent the trends in lipid class concentrations in HD and MA patients. The boxes represent data obtained in the range 25th–75th percentile; the line across the boxes indicates the median value; the lines above and below the boxes indicate extreme values (10th or 90th percentile). If not specified with fatty acid composition (e.g., 24:1, 18:0, 16:0, 64:8), the boxplot refers to the lipid class considering discriminant lipids only. Data were reported as log-transformed and auto-scaled mass intensities (Intensity norm). Outliers are displayed as separate points. Statistical significance was evaluated by unpaired *t*-test. *p* values are schematized as follows: ** < 0.01; *** < 0.001; **** < 0.0001.

**Figure 4 ijms-22-13410-f004:**
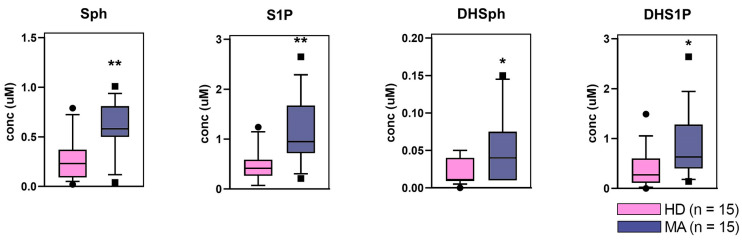
Quantitative evaluation of plasma sphingoid bases in MA patients (MA, n = 15) in comparison to age and sex matched healthy donors (HD, n = 15). Statistical significance was evaluated by unpaired *t*-test. *p* values are schematized as follows: * < 0.05; ** < 0.01.

**Figure 5 ijms-22-13410-f005:**
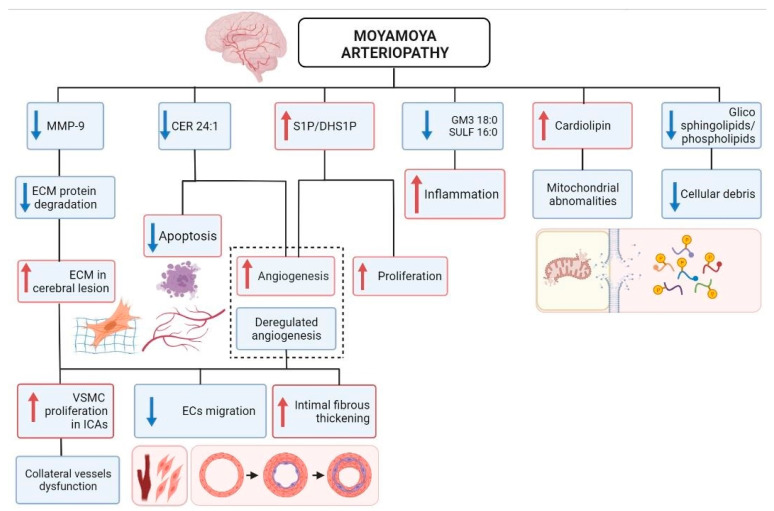
Map representing the main deregulated protein and lipid factors in MA plasma and the predicted associated biological effects. Red arrows indicate up-regulation of lipids or induction of associated biological effects; blue arrows indicate down-regulation of proteins/lipids or decrease of associated biological effects. Dotted line indicates the effects of reported deregulations on angiogenesis.

**Table 1 ijms-22-13410-t001:** Nomenclature of lipid species in MA plasma lipidomics.

CER 24:1	Ceramide 24:1	LPC	Lysophosphatidylcholine
CL 64:8	Cardiolipin 64:8	LPE	Lysophosphatidylethanolamine
DAG	Diacylglycerol	PC	Phosphatidylcholine
DHS1P	Dihydrosphingosine-1-phosphate	PE	Phosphatidylethanolamine
DHSph	Dihydrosphingosine	PI	Phosphatidylinositol
EtherPC	Phosphatidylcholine Ether	S1P	Sphingosine-1-phosphate
EtherPE	Phosphatidylethanolamine Ether	SM	Sphingomyelin
Gb3	Globotriaosylceramide	Sph	Sphingosine
GM3	Ganglioside GM3 18:0	SULF 16:0	Sulfatide 16:0
HEXCER	Hexosylceramide	TAG	Triacylglycerol
LACCER	Lactosylceramide		

## Data Availability

Data supporting reported results can be found in publicly archived datasets generated during the study at the Fondazione IRCCS Istituto Neurologico “C. Besta” (https://zenodo.org/communities/besta/?page=1&size=20; accessed on 15 November 2021).

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
