# Peer review of "Plasma Lipid Profiling Contributes to Untangle the Complexity of Moyamoya Arteriopathy"

_ijms, 2021, doi:10.3390/ijms222413410_

Round 1
Reviewer 1 Report
In this manuscript entitled “Plasma lipid profiling contributes to untangle the complexity of Moyamoya arteriopathy pathogenesis,” the authors evaluated the lipid profile of plasma of Moyamoya arteriopathy (MA) and found its change. This manuscript includes interesting observations; however, several descriptions were inadequate and included overinterpretations. My major comments are as follows: Major comments: 1. The authors determined the change of lipid profile and protein levels; however, the authors did not investigate the relationship between these changes and MA pathogenesis in the present study. Thus, the title of this study includes an overinterpretation of results. 2. In the introduction, the description of lipid mediators is less. Please describe it for the readers to understand this manuscript precisely. 3. Please adjust the format of figure 1 to other figures such as figure 2. 4. The authors identified the significant changes of several lipid contents in plasma of MA patients. Is there any correlation between these and pathological conditions? 5. Figures 5-7 are interesting but it appears to be a review paper. The authors should consider this.
Author Response
Reviewer 1
Comments and Suggestions for Authors
In this manuscript entitled “Plasma lipid profiling contributes to untangle the complexity of Moyamoya arteriopathy pathogenesis,” the authors evaluated the lipid profile of plasma of Moyamoya arteriopathy (MA) and found its change. This manuscript includes interesting observations; however, several descriptions were inadequate and included overinterpretations.
We are grateful to the Reviewer for his/her comments and for the careful revision. We are confident that the new information and additional sections that we added, together with the requested changes included will be convincing and will fill the gaps highlighted by the Referee.
My major comments are as follows:
- The authors determined the change of lipid profile and protein levels; however, the authors did not investigate the relationship between these changes and MA pathogenesis in the present study. Thus, the title of this study includes an overinterpretation of results.
We thank the Referee for this comment that evidenced a limit of our work and pointed out the more general and still open question about the pathogenesis of such a rare and still obscure cerebrovascular condition. We hope for future improvements associated to therapeutic application of the potential biomarkers identified here. However, we agree with the Referee and appreciate the suggestion, thus we have now removed the term “pathogenesis” from the Title (Plasma lipid profiling contributes to untangle the complexity of Moyamoya arteriopathy, see on page 1, line 3).
- In the Introduction, the description of lipid mediators is less. Please describe it for the readers to understand this manuscript precisely.
We agree with the Reviewer that not all potential readers may be experts in lipids and lipidomics, so we have added a section for that purpose in the Introduction, on page 2, lines 64-90. We have inserted some new refs (from ref 20 to ref 32), and accordingly renumbered the corresponding references and the Bibliography on page 16-17.
- Please adjust the format of figure 1 to other figures such as figure 2.
As rightly requested by the Reviewer, the format of Figure 1 has been completely revised and standardized to that of the following Figures (2-4). Figure 1 caption has been accordingly conformed to the new format (page 3, lines 121-127).
- The authors identified the significant changes of several lipid contents in plasma of MA patients. Is there any correlation between these and pathological conditions?
We thank the Reviewer for this useful comment/question and we better detailed the pathological characteristics of Moyamoya arteriopathy patients, whose plasma was analyzed using the lipidomic approach. To this aim, we have added a Supplementary Table S2 (see below and attached file) that includes the demographic, clinical and neuro-radiological features of such selected group of MA patients, to help readers in interpreting the results presented in Figures 2-4. The Table S2 was cited in Discussion, on page 8, line 323. Specifically, we reported the results of a homogeneous group of patients that is highly representative of the whole cohort of our patient. All of them were women with mean age around 45y, bilateral involvement and high disease severity, as assessed by Suzuki grading ≥ III. Given the tight homogeneity of our sample, we believe that our plasmatic lipid profile findings are likely associated with typical MA pathological features. Also considering the importance of the issue raised by the Referee, we included these considerations in the main text, specifically in Discussion, on page 8, lines 320-325.
|
ID |
Age |
Gender |
CVD type |
NIHSS |
U-B |
Suzuki grading |
Vascular risk factors |
Pharmacological therapy |
|
|
MA |
5 |
44 |
F |
HS |
5 |
B |
IV |
H |
AE, AD |
|
MA |
15 |
47 |
F |
IS |
8 |
B |
IV |
H, PI, |
AG, ASA, ST, AE |
|
MA |
19 |
42 |
F |
IS |
2 |
B |
IV |
HT |
AG, ASA, ST, AE |
|
MA |
21 |
50 |
F |
HS |
1 |
B |
IV |
- |
AD |
|
MA |
24 |
51 |
F |
TIA |
0 |
B |
III |
HoS, ET |
AG, ASA, ST, AE |
|
MA |
34 |
22 |
F |
TIA |
10 |
B |
IV |
H |
- |
|
MA |
40 |
47 |
F |
TIA |
3 |
B |
IV |
H, HoS, PI |
AG, ASA, Other |
|
MA |
41 |
52 |
F |
Other |
0 |
B |
V |
PI |
- |
|
MA |
42 |
39 |
F |
HS |
0 |
B |
III |
DL,HoS, PI |
ST |
|
MA |
43 |
45 |
F |
IS |
0 |
B |
IV |
PI, HHC |
AG,ASA |
|
MA |
45 |
60 |
F |
IS |
4 |
B |
V |
H, DL |
AG, ASA ST |
|
MA |
51 |
56 |
F |
IS |
0 |
B |
III |
H, DL, HoS, HHC |
AG, ASA, ST |
|
MA |
52 |
37 |
F |
IS |
2 |
B |
VI |
H, DL, ET. HT |
AG, ASA, ST, Other |
|
MA |
59 |
56 |
F |
TIA |
0 |
B |
IV-V |
H, DL |
AG, ASA, ST, AE |
|
MA |
61 |
38 |
F |
TIA,Other |
0-2 |
B |
IV |
HoS,PI |
AG, ASA, Other |
Supplementary Table S2: Demographic, clinical and neuroradiological features of 15 MA patients whose plasma samples were included in lipidomic analyses (AA, alcohol abuse; AD, antidepressant; AE, antiepileptic; AG, antiaggregants; AH, antihypertensive; AHHC, anti-hyperhomocysteinemia; ASA, acetylsalicylic acid; B, bilateral; CVD, cerebrovascular disease; DL, dyslipidemia; DM, diabetes mellitus; ET, estroprogestinic therapy; F, female; H, hypertension; HHC, hyperhomocysteinemia; HoS, history of smoking; HS, hemorrhagic stroke; HT, head trauma; IHH, ischemic heart disease; IS, ischemic stroke; M, male; MA, moyamoya arteriopathy; NIHSS, National Institute of Health scale; PI, physical inactivity; PSY, psychiatric disorder; ST, statins; TIA, transient ischemic attack; U, unilateral).
- Figures 5-7 are interesting but it appears to be a review paper. The authors should consider this.
We thank the Reviewer for this very useful suggestion. We have considered of moving Figures 5 (Implications of RNF213 in vasculogenic/angiogenic/inflammatory pathways) and 6 (Putative effects of RNF213 in lipid metabolism) into Supplementary Materials (as Supplementary Figures S1 and S2), as they concern more general aspects inherent to RNF213, and not to the results of the present study. Hoping to meet the approval of the Reviewer, we also believed that previous Figure 7 (Map representing the main deregulated protein and lipid factors in MA plasma and the predicted associated biological effects), which represents a summary scheme of the original results obtained in our study, can be kept in the main text. In order to help the interpretation of the potential origin of the reported plasma lipid and protein changes in a pathologically meaningful context, we mapped in the present Figure 5 (previous Figure 7) the deregulated species, as well as the predicted associated effects.
We hope that the Reviewer agrees with the interpretation of her/his comment. The text and the Figure citations have been accordingly modified, specifically on page 7, line 236 and line 242.

Reviewer 2 Report
Manuscript (ijms-1491031) titled: "Plasma lipid profiling contributes to untangle the complexity of Moyamoya arteriopathy pathogenesis" and submitted by Michele Dei Cas et al. to IJMS is very well written and described very important topic in edge and merging of two fields: analytical chemistry (Plasma lipid profiling using ELISA and mass spectrometry) and molecular sciences / clinical biochemistry (the complexity of Moyamoya arteriopathy pathogenesis). Moyamoya arteriopathy (MA) is a rare cerebrovascular disorder characterized by ischemic/hemorrhagic strokes. The main aim of authors was to evaluate angiogenic/vasculogenic/inflammatory proteins and lipid profile in plasma of MA patients and control subjects
(healthy donors HD or subjects with atherosclerotic cerebrovascular disease ACVD) and for this purpose authors applied ELISA and a complete lipidomic analysis they performed on plasma by mass spectrometry. According authors statement , their findings indicate that the plasma lipid profile of MA patients is definitely peculiar, thus highlighting a novel source of reliable clinically useful biomarkers for the disease.
In my opinion manuscript is very carefully written, authors put a lot of effort from basics (design of experimental part of study, collection of samples from patients etc) until comprehensive discussion and conclusions. If others reviewers and editor agree, could be accepted in present form.
Author Response
Reviewer 2
Comments and Suggestions for Authors
Manuscript (ijms-1491031) titled: "Plasma lipid profiling contributes to untangle the complexity of Moyamoya arteriopathy pathogenesis" and submitted by Michele Dei Cas et al. to IJMS is very well written and described very important topic in edge and merging of two fields: analytical chemistry (Plasma lipid profiling using ELISA and mass spectrometry) and molecular sciences / clinical biochemistry (the complexity of Moyamoya arteriopathy pathogenesis). Moyamoya arteriopathy (MA) is a rare cerebrovascular disorder characterized by ischemic/hemorrhagic strokes. The main aim of authors was to evaluate angiogenic/vasculogenic/inflammatory proteins and lipid profile in plasma of MA patients and control subjects (healthy donors HD or subjects with atherosclerotic cerebrovascular disease ACVD) and for this purpose authors applied ELISA and a complete lipidomic analysis they performed on plasma by mass spectrometry. According authors statement, their findings indicate that the plasma lipid profile of MA patients is definitely peculiar, thus highlighting a novel source of reliable clinically useful biomarkers for the disease.
In my opinion manuscript is very carefully written, authors put a lot of effort from basics (design of experimental part of study, collection of samples from patients etc) until comprehensive discussion and conclusions. If others reviewers and editor agree, could be accepted in present form.
We are very grateful to the Reviewer for her/his comments and kind appreciation for the experimental design of our study and for the final editing of our manuscript.
Round 2
Reviewer 1 Report
Thank you for your efforts.